# Cellular and Molecular Mechanisms Underlying Glioblastoma and Zebrafish Models for the Discovery of New Treatments

**DOI:** 10.3390/cancers13051087

**Published:** 2021-03-03

**Authors:** Pedro Reimunde, Alba Pensado-López, Martín Carreira Crende, Vanesa Lombao Iglesias, Laura Sánchez, Marta Torrecilla-Parra, Cristina M. Ramírez, Clément Anfray, Fernando Torres Andón

**Affiliations:** 1Department of Medicine, Campus de Oza, Universidade da Coruña, 15006 A Coruña, Spain; 2Department of Neurosurgery, Hospital Universitario Lucus Augusti, 27003 Lugo, Spain; 3Department of Zoology, Genetics and Physical Anthropology, Campus de Lugo, Universidade de Santiago de Compostela, 27002 Lugo, Spain; alba.pensado.lopez@rai.usc.es (A.P.-L.); martin.carreira@rai.usc.es (M.C.C.); vanesa.lombao.iglesias@sergas.es (V.L.I.); lauraelena.sanchez@usc.es (L.S.); 4Center for Research in Molecular Medicine and Chronic Diseases (CiMUS), Universidade de Santiago de Compostela, 15706 Santiago de Compostela, Spain; 5IMDEA Research Institute of Food and Health Sciences, 28049 Madrid, Spain; marta.torrecilla@imdea.org (M.T.-P.); cristina.ramirez@imdea.org (C.M.R.); 6IRCCS Istituto Clinico Humanitas, Via A. Manzoni 56, 20089 Rozzano, Milan, Italy; clement.anfray@humanitasresearch.it

**Keywords:** glioblastoma, cancer, tumor microenvironment, glioma-associated microglia/macrophages, genetics, metabolism, miRNA, zebrafish, drug discovery

## Abstract

**Simple Summary:**

Glioblastoma (GBM) is one of the greatest challenges facing neuro-oncology today. Current treatments are far from satisfactory and, given the poor prognosis of the disease, therapeutic efforts are focused on palliative management rather than curative intervention. Here, we review the cellular heterogeneity of GBM, including tumor cells and microglia/macrophages among others, as well as the genetic, epigenetic and metabolic alterations controlling its initiation and progression. Then, we describe the genetic and xenotransplantation zebrafish models established in the last few years for the study of GBM physiopathology and for testing new drugs to improve the treatment of the disease. Taking this information into account, forthcoming studies using zebrafish models of GBM are expected to shed light on better diagnosis and treatments, thus providing hope for GBM patients.

**Abstract:**

Glioblastoma (GBM) is the most common of all brain malignant tumors; it displays a median survival of 14.6 months with current complete standard treatment. High heterogeneity, aggressive and invasive behavior, the impossibility of completing tumor resection, limitations for drug administration and therapeutic resistance to current treatments are the main problems presented by this pathology. In recent years, our knowledge of GBM physiopathology has advanced significantly, generating relevant information on the cellular heterogeneity of GBM tumors, including cancer and immune cells such as macrophages/microglia, genetic, epigenetic and metabolic alterations, comprising changes in miRNA expression. In this scenario, the zebrafish has arisen as a promising animal model to progress further due to its unique characteristics, such as transparency, ease of genetic manipulation, ethical and economic advantages and also conservation of the major brain regions and blood–brain–barrier (BBB) which are similar to a human structure. A few papers described in this review, using genetic and xenotransplantation zebrafish models have been used to study GBM as well as to test the anti-tumoral efficacy of new drugs, their ability to interact with target cells, modulate the tumor microenvironment, cross the BBB and/or their toxicity. Prospective studies following these lines of research may lead to a better diagnosis, prognosis and treatment of patients with GBM.

## 1. Introduction

Glioblastoma (GBM) is one of the greatest challenges facing neuro-oncology today. This disease represents the most common of all brain malignant tumors, comprising more than 50% of existing high-grade gliomas. GBM has an annual global incidence of 3–5 cases per 100,000 inhabitants and a slight predominance in males. This incurable malignant tumor, with almost non-existent long-term survivors, can occur at any age but presents a clear peak with the highest incidence in the sixth decade of life [1,2].

Most GBM arise de novo, while secondary GBM tumors commonly develop from lower grade gliomas [1]. The complexity of managing GBM patients depends on many factors, including tumor size and location, age, the Karnofsky Performance Scale Index, tumor histology and the status of molecular markers. The diagnosis of GBM is based on histological techniques, with a high-grade glioma being defined by atypia, cellular pleomorphism, mitosis, vascular proliferation and necrosis, according to the criteria established by the World Health Organization (WHO) classification. By definition, GBM corresponds to a grade IV and has the worst prognosis among infiltrating gliomas [3]. In clinical practice, isocytrate dehydrogenase (IDH) mutations and methylation of the promoter O(6)-methyl guanine methyl transferase (MGMT) gene are routinely evaluated to improve diagnosis and the classification of tumors and to estimate the sensitivity of the tumor to alkylating agents such as temozolomide (TMZ) [4,5]. Despite this, the aggressive and invasive development and growth of brain tumors, the impossibility of completing tumor resection, the limitations for drug administration and therapeutic resistance to treatment are the main problems presented by this pathology. While overall survival after surgical resection of the tumor is 3 to 6 months, the inclusion of radiotherapy in the treatment plan increases this parameter to 12.1 months (2-year survival of 10.4%) and a slight increase in survival to 14.6 months (2-year survival 26.5%) can be achieved by the addition of concomitant and adjuvant chemotherapy with TMZ. In addition, current treatments are far from satisfactory and, given the poor prognosis of the disease, therapeutic efforts are mainly focused on palliative management rather than curative intervention.

In this scenario, different promising lines of research have been initiated for the development of new therapeutic strategies to treat GBM [2]. From a cellular and molecular point of view, GBM tumors present high heterogeneity [6,7,8,9], which contributes to their recurrence and therapeutic resistance [10,11]. In addition, understanding the tumor microenvironment (TME), the cellular origin and the molecular alterations of the tumor cells and immune related cells will potentially help to improve the diagnosis, prognosis and treatment of the disease.

The zebrafish (*Danio rerio*) has become a well-established model for studying the physiopathological features and screening of new treatments for several human diseases, including cancer. Rapid embryo development, its small size and its transparency, genetic and physiological conservation and ethical and economic advantages have made zebrafish stand out from all other in vivo models [12]. Concerning GBM, the simplicity to perform genetic manipulations and the transplantation of human tumor cells, together with the conservation of the major brain subdivisions in this species, enables researchers to recapitulate the characteristics of human tumors and their related TMEs, in turn, allowing the mechanism of action of new therapeutic strategies to be evaluated [13].

In this manuscript, we review the physiopathology of GBM from a cellular and molecular perspective, with a particular focus on macrophages/microglia in the TME, genetic, epigenetic and metabolic alterations and possibilities for their intervention by microRNA (miRNA) manipulation. Then, we provide our view on the opportunities and challenges of zebrafish models for improving our understanding of the disease and the evaluation of new treatments.

## 2. Cellular Pathology and Tumor Microenvironment in Glioblastoma: Macrophages/Microglia

It is desirable to understand the cellular origin of the disease and the composition of the TME to establish an early diagnosis, identify therapeutic targets and improve its outcome. The genomic, epigenomic and transcriptomic characterization of GBMs has provided information which allowed to establish the different tumor subtypes: proneural, mesenchymal and classical [14,15,16,17,18,19]. Aberrations in the expression of platelet-derived growth factor receptor alpha (PDGFRA), neurofibromatosis type I (NF1) and epidermal growth factor receptor (EGFR) were associated with the proneural, mesenchymal and classical subtypes, respectively [19]. However, these mutations can co-exist within a single tumor, both at the regional and single-cell levels, thus, in this case the designated subtypes reflect the dominant transcriptional program of a specific tumor within a particular time and space of sample isolation [8,18,19,20]. After oncogenic alterations, GBM cells can be differentiated and de-differentiated and can acquire stem cell properties [21,22]. All of this suggests that genetic, epigenetic and metabolic alterations (reviewed in Section 3) can be targeted, rather than simply killing a particular population of tumor cells, so are valuable strategies to treat the disease. In addition, to tumor cells, the therapeutic targeting of immune cells has gained high relevance in the last decade with the outstanding results achieved by anti-tumoral immunotherapy in other types of cancer (i.e., melanoma and lung cancer) [23].

The glioma tissue is infiltrated by many cells of different ontogenies, mostly resident microglia and tumor-infiltrating monocytes, which are denoted glioma-associated macrophages/microglia (GAMs) and which can represent from 30 to 50% of the total cells in the tumor [24,25,26]. Historically, macrophages and microglia had been thought to be transposable in the TME as they share immunologic functions such as phagocytosis or antigen presentation. However, in recent years, single-cell RNA sequencing (RNA-seq) performed on GAMs from human glioma tissues have identified two phenotypically distinct subsets associated with gene signatures of microglia-enriched and bone marrow-derived macrophages [27]. These findings have also revealed key differences in the way the glioma microenvironment shapes the transcriptional expression of these cells, in particular at the level of genes involved in the secretion of inflammatory cytokine and antigen presentation [28,29,30]. Moreover, fate mapping analysis has revealed that microglia cells derive from primitive myeloid progenitors which enter the brain during embryogenesis [31]. Conversely, circulating blood-monocytes typically enter the brain in pathological situations and they differentiate into macrophages [32]. The exact contribution of microglia and circulating blood-monocytes to the total pool of GAMs is still an open question. In some models, microglia predominance has been reported [33], whereas others have demonstrated that infiltrating monocytes are accountable for the majority of the GAM population [29,34]. Nevertheless, the key drivers of GAM recruitment are contextual, as these chemokines are heterogeneously expressed among glioma, probably reflecting the diversity in glioma genetics. Transcriptional classifications have revealed that this diversity influences the extent of GAM infiltration. For instance, phosphatase and tensin homolog (PTEN) deficiency in glioma drives macrophage infiltration via up-regulation of lysyl oxidase (LOX) [35] and in mesenchymal GBMs neurofibromin 1 (NF1) deficiency results in increased GAM infiltration compared with proneural or classical subtypes [18,36,37]. Notably, it is now recognized that microglia and macrophages colonize different regions of gliomas. Monocyte-derived macrophages are enriched in the tumor core where they occupy perivascular regions, while microglia-derived tumor associated macrophages are typically found at the tumor periphery [29,38]. Intravital 2-photon microscopy has revealed that these two subsets are also morphologically distinct, monocyte-derived cells, being small and highly migratory, while microglia are large, branched cells with highly active processes continuously extending and retracting within tumors [39]. Future studies will be necessary to investigate whether this divergence is related to a different activity and if these two populations affect tumor growth differently.

Although the mechanisms by which GAMs migrate to the tumor site are not entirely clear, some studies have shown that their recruitment is mediated by various glioma-derived factors such as CCL2, CX3CL1, SDF-1, CSF-1, GM-CSF and EGF, which act as chemoattractants for GAMs and mediate the crosstalk between tumor cells and the innate immune system [40] (Figure 1). In an NF1 mutant optic glioma murine model, Guo et al. showed that CX3CL1 is a key chemokine responsible for the attraction of microglia to the tumor [36]. Using CX3CR1- and CCR2-engineered murine models, Chen et al. demonstrated that in glioma, microglia only express CX3CR1, whereas most inflammatory monocytes/macrophages express both CX3CR1 and CCR2. This suggests that CCL2 is a major attractant for monocytes. Accordingly, in this model, CCL2 depletion led to pro-longed survival [29]. CCL2 produced by both glioma cells and GAMs, has also been shown to be essential for the recruitment of Treg cells and myeloid-derived suppressor cells [41], which act as drivers of the immune-suppressive phenotype typical of these tumors. Another recent study has suggested that osteopontin might also act as an important chemokine for macrophage recruitment to GBM tumors triggering its binding to integrin αvβ5 [42]. T-cell dysfunction is also a hallmark of high-grade glioma, reflecting the active immunosuppressive microenvironment which contributes to tumor immune escape in patients. Takenaka et al. showed that expression of CCR2 is promoted in GAMs, by the activation of the aryl hydrocarbon receptor (AHR), causing cytotoxic T-cell dysfunction through the CD39/CD73/adenosine pathway [43]. Anti-tumoral functions of tumor-infiltrating lymphocytes (TILs) are impaired by multiple immunosuppressive factors produced by GAMs and glioma cells and these factors are also more likely to up-regulate multiple immune checkpoint molecules such as PD-1, TIM-3 and LAG-3 [44].

Phenotypically, it is now well documented that GAMs do not fit with the classical M1/M2 macrophage dichotomy [45]. Originally suggested to be simplified as macrophages, the phenotypic dichotomy assigned to T-helper cells (Th1/Th2), the M1/M2 denominations of macrophages, were later described by Mantovani et al. as the two extreme poles of a continuum of phenotypes and functions which could be acquired by very plastic immune cells [46]. In this outline, classically activated M1-like macrophages present a prototypical pro-inflammatory response and anti-tumor functions, while alternatively, activated M2-like macrophages, resembling tumor associated macrophages (TAMs), present anti-inflammatory properties, at the same time as supporting tumor progression, angiogenesis and metastasis and last of all, prevent adaptive immune responses [47].

In the context of glioma, using genome-wide microarray analysis, Szulzewsky et al. revealed that a significant amount of genes up-regulated in GAMs do not fit with the M1/M2 classification [48] and the single cell RNA-seq demonstrated that GAMs express rather a mix of M1/M2 markers [27,49]. Despite such controversy, some typical M2 markers of macrophage polarization which favor tumor progression have also been found in glioma, some examples of which are provided below. Glioma heterogeneity may explain why currently, the correlation between TAM infiltration in glioma and patient outcome is still not clear. Some studies have reported on a positive correlation between patient outcomes and the presence of TAMs in the vital tumor core in IDH1R132H-non-mutant GBMs [49]. In another study, M2-like TAM numbers, identified with CD204, increased with malignancy grade and were associated with a poor prognosis, while high IBA-1 intensity, a marker of activated macrophages, correlated with a longer survival [50]. Similarly, compared with IDH-wildtype GBM, a reduced number of macrophages which were more oriented towards a pro-inflammatory M1-like activation state were found in IDH-mutant GBM patients, possibly contributing to their prolonged survival [51]. As a whole, to understand GAM biology in gliomas, it is critical to evaluate an integrated vision, including their ontogeny, genomic and phenotypic diversity, differential location in the TME and functional activity. Preclinical studies using animal models have shown that GAMs play a major role in gliomagenesis and sustain tumor growth in both low-grade and high-grade gliomas. In murine models, depletion of macrophages/microglia has led to a reduction in tumor growth [32] and in line with this, some factors released by GAMs such as IL-6, IL-1β, EGF, STI-1 and TGF- β can promote tumor growth [32]. Indeed, the role of microglia in glioma invasiveness has also been related to the secretion of SIP1, EGF and TGF-β [52]. Moreover, the activation of TLR2 expressed on microglia, has triggered the release of matrix metalloproteinases (MMP2 and MMP9) [53,54], degrading the extracellular matrix and facilitating the invasion of glioma cells. Toll-like receptor 4 (TLR4) signaling in microglia has been shown to induce the release of IL-6 [55], a ligand of the signal transducer and activator of the transcription 3 (STAT3) pathway in gliomas, which is also involved in invasiveness [56]. Up-regulation of TGF-β and M-CSFR in the context of hypoxia, the main feature of high-grade gliomas, has been shown to induce the M2-like protumoral polarization of GAMs in murine glioma models [57,58]. The administration of acriflavine, a hypoxia-inducible factor (HIF) inhibitor, hampered GAM enrichment and polarization and significantly inhibited tumor progression [38].

To date, the vast majority of studies have demonstrated that macrophages/microglia support GBM progression, although, in some cases, the ability of these cells to fight against the tumor has been reported. This work has also revealed some molecular targets offering possibilities for therapeutic intervention. Although, to our knowledge, studies testing macrophage-directed therapies to treat GBM have still not been performed in zebrafish, several investigations have already evaluated the role of macrophages and microglia in zebrafish regeneration, tumor origin and progression [59,60] and in the context of other diseases [61,62]. Relevant zebrafish models for the study of GBM and the evaluation of new treatments are reviewed in Section 4 and Section 5, respectively.

## 3. Molecular Pathology in Glioblastoma

### 3.1. Genetic Mutations in Glioblastoma

Three important genetic events drive the development of GBM: the dysregulation of growth factor signaling (EGF, PDGF, VEGF, etc.), the activation of the phosphatidylinositol-3-OH-kinase (PI(3)K) pathway and the inactivation of the p53 and retinoblastoma (Rb) tumor suppressor pathways [15]. IDH mutation occurs early in this process and stands out as the main biomarker used in the 2016 CNS WHO [3] to identify and classify GBM into the following types: (1) Glioblastoma IDH-wildtype (about 90% of cases): primary or de novo GBM that predominates in patients of over 55 years of age. (2) Glioblastoma IDH-mutant (about 10% of cases): secondary GBM that arises in younger patients. (3) Glioblastoma not otherwise specified (NOS), when there is no information on this gene. The mutation of arginine to histidine on codon 132 (p.R132H) is most frequent. In GBM, IDH1/2 mutations correlate with better prognosis, while no IDH3 mutations have been associated yet. IDH mutation is strongly associated with 1p/19q codeletion and MGMT promoter methylation, another common diagnostic biomarker, but are mutually exclusive with EGFR amplification (gain of chromosome 7) and PTEN deletion (loss of chromosome 10) [63]. While the methylation of the MGMT gene promoter helps to estimate the sensitivity of the tumor to alkylating agents such as TMZ, the relationship of IDH1/2 mutations with response to chemotherapy remains controversial [64].

As an example of the dysregulation of growth signaling commonly occurring in GBM, 57% of GBM shows evidence of gains in function mutation and/or focal amplification of EGFR, associated with an increase in the aggressiveness of these gliomas [14]. The most frequent EGFR mutation in GBM that occurs, EGFR^vIII^, contains an in-frame deletion within the extracellular domain that provides constitutive activation in a ligand-independent fashion and promotes cell proliferation via the Ras-MAPK and PI3K pathways. Deletion of PTEN displays a similar effect [65].

Cycle checkpoint proteins are often altered in cancer. p53 is a classic tumor suppressor that regulates many genes involved in the cell cycle and apoptosis cascades. Inactivation of p53 in GBM happens with a variety of mechanisms, including amplification of p53 inhibitors such as murine double minute (MDM) 2 and MDM4 (the latter appears most commonly in tumors with no TP53 or telomerase reverse transcriptase (TERT) mutations), deletion of p53 stabilizers such as p14/ARF and mutation in the TP53 gene which occurs in 85% of GBMs [14]. Cyclin-dependent kinase inhibitor 2A and B (CDKN2A/B) copy number losses (homozygous deletion of CDKN2A-p16^INK4α^ in chromosome 9p) are linked to the activation of the Rb pathway and to the proliferative niches that are observed in gliomas [66]. Other alterations are responsible for the progression and difficult treatment of GBM. Mutations in the promoter of the telomerase reverse transcriptase (TERTp) or ATRX chromatin remodeler (ATRX) (mutually exclusive since they have similar functions), lead to an increased lengthening of telomeres, allowing cells to overcome cellular senescence and promoting immortalization [67,68].

### 3.2. Epigenetic Alterations in Glioblastoma

Genetic mutations are not the sole participants in the development and progression of GBM. Epigenetics, as shown in many other types of cancer, plays an important role too. GBM commonly presents an extended genomic epigenetic hypomethylation that allows the transcription of multiple genes and is associated with rapid progression. Some of the regions affected include D4Z4 (a polymorphic repeat structure), oncogenic genomic loci such as SAT2 and the oncogene MAGE-A1 [69]. Mutations in the IDH1/2 genes induce the production of an alternative metabolite, 2-hydroxyglutarate (2-HG) at the expense of alpha ketoglutarate (α-KG) [70]. Since numerous enzymes involved in epigenetics are dependent on α-KG (such as histone demethylases or ten-eleven translocation (TET) enzymes), the absence of this compound inhibits them, increasing the methylation of CpG islands and histones. This intense methylome remodeling induces what is known as the glioma CpG island methylator (G-CIMP) phenotype [71] and contrasts with the general state of hypomethylation observed in GBM. The rare occurrence of IDH-mutation, a genetic marker of secondary GBM that is mostly absent in primary GBM and its correlation with better prognosis could indicate that this mutation is a hindrance in the progression from low grade glioma (LGG) to GBM since it changes the epigenetic state of multiple genes involved in important cellular processes. A glioma with IDH-mutation could require further alterations to develop a more aggressive phenotype than gliomas without it.

A key example that highlights the importance of epigenetics in GBM comes from MGMT alterations, one of the gold standard markers used for the characterization of the disease. The methylation of the MGMT promoter is a biomarker with great diagnostic value that correlates with good prognosis. Furthermore, as with the DNA alkylation repair enzyme, the inhibition of MGMT expression enhances the anti-tumoral efficacy of TMZ [72].

The epigenetic inactivation of many tumor suppressor genes has also been observed in GBM. PTEN, p53, RB, p14 and p16 appear hypermethylated, along with some other genes involved in key processes such as intercellular contact (PCDH-γ A11), the MAPK pathway (SOCS1) and apoptosis (caspase-8) [69]. Histone deacetylases (HDACs) have been implicated in many GBM tumors owing to the fact that their dysregulation influences tumor progression, invasion and resistance to therapy. This is of special relevance in GBM stem cells (GSC), where dysregulated HDAC expression has been associated with altered signaling mechanisms like the sonic hedgehog (SHH) pathway (essential for stemness, viability and radio-resistance) and the maintenance of mitochondrial functions and metabolic adaptions [73].

### 3.3. Metabolic Changes in Glioblastoma

Metabolic switching is a well-known hallmark of cancer, including GBM and understanding and reprogramming metabolic pathways in tumors may yield new treatment options [74]. Many cancers prefer aerobic glycolysis as their metabolic program of choice to fulfill their bioenergetic and anabolic requirements for rapid growth and enhance their survival in response to microenvironmental stress [75]. Aberrant expression of oncogenes and tumor suppressor genes in GBM has correlated with changes in the expression and activity of glycolytic transporters. Aerobic glycolysis, described in the 1920s by Otto Warburg and his colleagues, known as the Warburg Effect, is defined as an increase in the rate of glucose uptake and preferential production of lactate in the presence of oxygen in cancer cells in compensation for insufficient OXPHOS [76,77]. The advantage that the Warburg Effect confers to cancer cells is not completely clear. However, since ATP levels do not seem to represent a limiting factor for tumor cells, it is generally accepted that aerobic glycolysis in cancer promotes the use of NADH as a by-product of lactate to generate biomass and lactate to acidify the microenvironment, facilitating tumor invasion [78]. Aerobic glycolysis, which favors the deadly progression of GBM [79] (Figure 2), is controlled by glucose transporters, GLUT1-4 and key glycolytic enzymes such as these: HK1-3, PFK1, GAPDH, PKM2 and LDHA; they are also influenced by several other cellular pathways and regulatory proteins, including the following: PI3K/AKT, LKB1/AMPK, HIF-1/2, p53, EGFR, PDGFR and c-MYC, among others. Data obtained from The Cancer Genome Atlas (TCGA) have identified three core pathways which are frequently altered in more than 75% of GBMs: (i) receptor tyrosine kinase/RAS/phosphatidylinositol 3 kinase (RTK/RAS/PI3K) signaling (ii) p53 (iii) Rb signaling networks [15]. The analysis of 96 GBM human samples showed aberrant overactivation in a number of RTKs and gain of function mutations such as in EGFR (observed in 45% of GBM cases) which triggers RAS and AKT/PI3K cascades. As a consequence, these downstream proliferative and metabolic signaling cascades have been found to be up-regulated in GBM [15]. The activation of the mTOR signaling cascade by AKT in GBM has led to the up-regulation of transcription factors such c-Myc [37], which up-regulates the expression of glycolytic genes [80]. AKT can also up-regulate glycolysis by activating transcription and translocation of glucose transporters (GLUTs), also up-regulated in GBM and other key enzymes such as Hexokinase II, involved in the first step of glycolysis [81]. Therefore, alterations in oncogenes and tumor suppressor genes controlling metabolism and metabolic enzymes underlying tumor progression in GBM are targets of interest for the treatment of the disease. 

It is of note, that mitochondria are a key cellular organelle regulating energy metabolism, generation of free radicals and apoptosis. Additional metabolic signs of malignancy in GBM have been linked to mitochondrial dysfunction and morphological abnormalities, which have been correlated with compromised energy/ATP production via OXPHOS and reduced apoptosis. Impaired mitochondrial metabolic capacity in glioma cells has been evident from the identification of mutations in gene coding for IDH, described in Section 3.1 and a principal component of the Krebs cycle. Mutations in the NADPH-linked mitochondrial isoforms of IDH1 and IDH2 have led to impaired energy production in the mitochondria and thus, provide clear evidence for mitochondrial dysfunction in gliomas [82]. Indeed, IDH1 mutation not only causes 2-HB build-up, but also broad changes in cellular metabolism. Due to changes in the expression of several key enzymes, GBMs containing IDH1 mutations have been shown to exhibit a reduced glycolytic phenotype compared to GBM without the mutation. Therefore, IDH status can also indicate distinct glycolytic phenotypes of GBM, which may contribute to different clinical behavior of tumors with and without the IDH1 mutations [80]. Cardiolipin, an important phospholipid concentrated at the contact sites of outer and inner mitochondrial membranes and at the electron transport chain (ETC) associates with respirasome complexes that are crucial for mitochondrial function. Evidence from experimental and clinical studies has shown defects in mitochondrial ETC associated with cardiolipin biosynthesis in gliomas. Thus, the degree of faulty cardiolipin synthesis and its function indirectly affects the mitochondrial metabolic capacity and intrinsic apoptosis execution which have been found to be impaired/malfunctioned in certain grades of glioma cells [82].

In addition, to genetic and epigenetic intervention, a broad variety of biological molecules have been evaluated for the therapeutic manipulation of metabolic pathways, including sugars, lipids and proteins [83]. Furthermore, with the same purpose, novel post-transcriptional regulators, such as miRNAs (reviewed below), are of utmost interest.

### 3.4. miRNAs in Glioblastoma

miRNAs are small (18–25 nucleotides), evolutionarily conserved, non-coding RNAs with important functions in gene regulation, acting predominantly at the posttranscriptional level. miRNAs are transcribed in the nucleus by RNA polymerase II into primary transcripts (pri-miRNAs) that are then processed sequentially in the nucleus and cytoplasm by a complex of RNase III-endonucleases, namely Drosha and Dicer [73,84,85], (Figure 2). Drosha, specifically, processes the pri-miRNA transcript to a 70–100 nucleotide stem-loop precursor (pre-miRNA), which is then delivered to the cytoplasm by Exportin 5, where it is subsequently cleaved by Dicer to produce a miRNA duplex [73,84]. The resulting duplex is then incorporated into the RNA-induced silencing complex (RISC) in association with an Ago family member. One of the strands (the passenger strand) is degraded, while the other strand (the mature miRNA) remains associated with the Ago protein and binds to partially complementary sites in mRNAs. By way of binding to the 3′UTR of target messenger RNAs (mRNAs), miRNAs repress translation or induce mRNA degradation [86,87].

Over the past two decades, miRNAs have been shown to regulate many physiological processes in the cells and they have been implicated in a wide range of pathological conditions, including cancer. Around 50% of the human-miRNA-encoded genes are located in genetic loci associated with cancer [88] and nearly all tumors show dysregulated miRNA expression signatures [89]. These alterations frequently correlate with up-regulation of oncogenes and/or downregulation of tumor suppressor-genes, therefore, promoting the development of the tumor. The pattern of miRNA expression is now being used, together with gene-expression profiling to stratify GBM patients into different groups [90]. In addition, miRNAs have multiple targets, allowing them to modulate many pathological aspects critical to cancer progression, including proliferation, cell death, metastasis, angiogenesis and drug resistance. A number of miRNAs have been identified that interact with known genes and pathways that underlie GBM [91] and some of them have been placed as biomarkers [92] and predictors of survival and outcomes [93].

The role of different miRNAs in regulating classic pathways and genes critical for the GBM genesis at the post-transcriptional level has been described. These include p53, EGFR, PDGFR, PTEN, PI3K and AKT and MGMT, among others [94] (Figure 2). MicroRNA signatures with deficient patterns of expression have been found for several miRNAs such as the miR-7, miR-34a, miR-128, miR-124, miR-137 and miR-181 families and it has been observed that their overexpression negatively impacts GBM development [95]. On the contrary, multiple up-regulated miRNAs have been described, miR-21 being the first-investigated oncomiR which plays crucial roles in deleterious processes in GBM by targeting aforementioned genes, as well as others involved in proliferation, cell survival and drug resistance [95]. Other examples of miRNA up-regulated in GBM with important functions in gliomagenesis include the miR-17-92 cluster, miR-10b, miR-15b, miR-26a, miR-93, miR-148, miR-182 and miR-221/222 [95]. In addition, to specific tissue-signatures associated with GBM tumors, miRNAs have been found in extracellular vesicles from GBM in circulation. This finding has encouraged studies to explore their use as potential biomarkers with diagnostic value by non-invasive techniques, such as liquid biopsies from GBM patients [94,96]. Along with the modulation of oncogenic signaling pathways and tumor suppressors, miRNAs have also been shown to govern the expression of key metabolic genes in the context of many diseases, as well as in GBM [80,97,98,99] (Figure 2). Thus, by targeting key proteins or enzymes that participate in processes like glycolysis, oxidative phosphorylation, glutamine, glucose and lipid metabolism, several microRNAs can contribute to the characteristic metabolic switching in cancer and GBM [85]. Some examples affecting glycolysis include miR-106a and miR-143 which targets the GLUT-3 transporter, as well as miR-326 targeting HK-2 and let-7a, which represses PKM2 expression. Other miRNAs interfere with mitochondrial function and energy metabolism (i.e., let-7, miR-16 and miR-23) or regulate mitochondrial proteins (i.e., ATP5A1 and ATP5B) in GBM. As aerobic glycolysis is a hallmark of GBM tumors, the ability of certain miRNAs to regulate glycolytic metabolism in GBM via regulating oncogenes and tumor suppressor genes in the RTK pathways and their downstream effector pathways such as the PI3K/AKT pathway, is of interest for therapeutic purposes [100]. Undoubtedly, further research efforts will elucidate novel roles of miRNAs and other posttranscriptional regulators in energy metabolism, providing new potential targets for the treatment of GBM and other diseases. To our knowledge, no studies with therapeutic purposes have yet been performed which manipulate miRNAs in GBM zebrafish models.

Overall, this knowledge and further understanding of molecular mechanisms controlling GBM pathology will offer numerous possibilities for therapeutic manipulation of signaling pathways, often intertwined, which can theoretically be treated with different drugs at the genetic, epigenetic or metabolic level. Genetic or xenotransplantation models of zebrafish selectively manipulated in these key pathways are particularly suited to the testing of new therapeutic strategies. In this context, miRNAs represent an attractive target which can be efficiently blocked by (i) sequence-specific oligonucleotides, (ii) antisense approaches or (iii) miRNA sponges.

## 4. Zebrafish Models of Glioblastoma

### 4.1. Advantages and Limitations of Current Zebrafish Models for Modelling Human Disease

Over the last few decades, advances in biotechnology have allowed improved in vitro and in vivo models of human disease to be developed, underlying pathogenic mechanisms to be understood and therapeutic approaches to be improved. Although in vitro systems for cancer research have experienced significant advances, living organisms bearing tumors are still necessary to recreate the “atmosphere” of human tumors; such examples could be the interaction between tumor cells and immune cells in the primary tumor or at metastatic sites, endocrine or metabolic factors altering tumor development, not to mention biodistribution, pharmacokinetics or efficacy of new drugs [101,102]. Despite rodents, mainly mice, being the most commonly used animal model for in vivo studies, zebrafish models are increasingly being used for cancer and drug discovery research. Since the early 1960s, when it was introduced as an animal model to study vertebrate developmental biology and genetics, zebrafish has progressively been adopted as a useful model for unraveling the cellular, molecular and genetic features of several human diseases, including cancer [12].

The success of zebrafish in biomedical research can be explained by its inherent biological features and economic issues. Zebrafish are able to produce between 200 and 400 embryos per couple once a week and their small size allows handling large numbers of fish in reduced space. Thus, zebrafish models enable high-throughput assays with significant statistical power, cost-efficient experiments and fewer ethical issues than murine models. The optical transparency of embryos allows direct visualization of development and makes the imaging and tracking of injected tumor cells possible without the need of invasive techniques [103]. Its translucent nature also enables the establishment of different cell or tissue-specific reporter lines.

Zebrafish, or murine, xenograft tumor models are commonly used in cancer research and are very useful for testing new drugs. However, these models present limitations related to the lack of a fully functional immune system as required to favor the implantation and growth of tumor cells in other species. Zebrafish complete their basic development in the first 24 h post-fertilization (hpf) but they lack an adaptive immune system until the first 12 to 14 days of development, thus allowing this window for the implantation and progression of the cancer cells, without the need to use immunocompromised fish [104]. Despite the peculiarity of the species initially being considered a limitation, recent studies have used these features as an opportunity for rigorous experiments, with some strategies being implemented for research on immune cells. Appropriate zebrafish models are routinely used for studies of innate immunity, using fluorescently labeled macrophages or neutrophils (i.e., GFP or RFP) as these cells are already available from the first stages of zebrafish development. As a solution for studying adaptive immunity, the development of human T cell tumor-implanted zebrafish models and the subsequent eradication of cancer cells have proven to be useful for precision medicine platforms when assessing the anticancer effects of cellular immunotherapy in vivo [105]. 

Nevertheless, xenografts are not applicable for every tumor type. Due to the phylogenetic distance between teleost fish and mammals, not all human organs are present in zebrafish. In order to overcome this hurdle, it has been suggested that analogous structures, such as gills could be used as a substitute for lungs [106]. With respect to brain tumor modeling, despite the differences between zebrafish and humans, it is worth noting that the most relevant brain regions and subdivisions, as well as cell types, differentiation, connectivity, signaling pathways and gene expression patterns, are highly conserved [13,107]. The immaturity of certain tissues in zebrafish embryos (i.e., myelinated axonal sheaths) does have to be considered though, as this may affect the development of certain tumors [108,109].

Concerning genetics, the release of the sequence assembly of the zebrafish genome has revealed that around 70% of human genes are shared in zebrafish with more than 80% of them being human disease-related genes [110]. Moreover, the relative cost-effectiveness and ease to perform genetic manipulations in zebrafish embryos [111] increases its attractiveness as a form of generating genetic cancer models. On the other hand, some critical oncogenic factors such as INK4^α^/ARF are not expressed in zebrafish which rules out studies related to these pathways [112]. Similar issues might be encountered for epigenetic or metabolic alterations related to cancer in zebrafish, but the amount of information in this regard is still very limited and has been reviewed by others [102,113].

Taking these pros and cons into consideration, below we describe relevant genetic and transplantation strategies that have been investigated using zebrafish as a model for GBM.

### 4.2. Genetic Zebrafish Models of Glioblastoma

Genetic approaches for disease modeling in zebrafish comprise several strategies ranging from gene-targeted mutations to transient or stable over-/downregulation of genes of interest. Numerous cancer models have been established by transgenic expression of human and murine oncogenes in zebrafish (i.e., KRAS^G12V^, HRAS^G12V^ and BRAF^V600E^) providing information about the role of selected genes in tumorigenesis [114,115,116]. Furthermore, zebrafish cell- and tissue-specific reporter lines have been developed to study cancer-related properties such as tumor cell growth, migration, invasion, angiogenesis, drug responses and interactions with neural and immune cells. For instance, Tg(*mpx*:GFP) for neutrophils [117] and Tg(*mpeg1*:EGFP) or Tg(*mpeg1*:mCherry) for macrophages [118,119] are frequently used. Among different methodologies for genetic manipulation, the Gal4/UAS system has been widely used for the activation of gene expression [120]. Other genetic models which could be useful for research in cancer or other diseases have recently been reviewed by Pensado-López et al. [107]. 

To model GBM-like tumors in zebrafish, several genetic approaches have been used (Table 1). To illustrate this, Ju et al. used a binary transgenic approach co-expressing zebrafish Smoa1 involved in the Shh pathway, with human AKT1 so as to model GBM-like tumors in the brain, retina and spinal cord [121]. The same group also developed the first animal model of gliomagenesis by Shh activation in neural progenitor cells by way of the transgenic expression of Smoa1 under the *krt5* neural promoter [122]. In zebrafish, KRAS^G12V^ overexpression under the neural promoters *krt5* and *gfap* (specific for neurons, glial cells and astrocytes) led to malignant tumors in the cranial cavity and parenchymal brain tumors respectively, highlighting that different tumor initiating cells may well determine the tumor type and demonstrating that tumorigenesis is driven by the activation of the canonical Ras and mTOR pathways [123]. Along the same lines, Mayrhofer et al. observed brain tumor development with the expression of HRAS^G12V^, YAP^S5A^, KRAS^G12V^, AKT, EGFR^vIII^ and BRAF^V600E^ oncogenes under the control of the zic4 enhancer, proliferating domain of developing nervous system. The somatic expression of activated RAS promoted brain tumors and/or heterotopia, suggesting their possible origin from a benign developmental lesion. Analyses of global RNA expression revealed that developed tumors resembled mesenchymal GBMs with a strong YAP component, suggesting YAP as a hallmark of malignant brain tumor and providing a useful model for preclinical drug screening for this GBM subtype [124]. As somatic missense mutations in the IDH1 gene are frequently found in gliomas and have been related with better patient survival, several transgenic zebrafish lines expressing various IDH1 mutations have been established to clarify their role in tumor development [125].

Transient genetic modifications in zebrafish are commonly performed by the morpholino-based (MO) expression silencing, which enables the binding of the oligonucleotide to a desired target, knocking down the gene without sequence modification [126]. MOs have been extensively used to unravel the basis of GBM-cells, to elucidate molecular mechanisms and to find novel prognosis markers and therapeutic strategies. As an example of this, experiments knocking down Ephrin-B3 ortholog and co-silencing EphA4 in zebrafish allowed impairments to be found in the formation of intersegmental vessels (ISVs) associated with EphA4-related apoptosis and angiogenesis, thus suggesting EphA4-induced cell death as an opportunity to slow GBM growth [127]. Similarly, the involvement of Plexin-A1 in the development of blood vessels and angiogenic sprouts in ISVs was demonstrated by Plexin-A1 knock down in zebrafish models [128].

**Table 1 cancers-13-01087-t001:** Genetic approaches using zebrafish for modeling GBM.

Genetic Approach	System	Gene/Protein	Zebrafish Strain	Notable Results	Reference
Transgenesis	Gal4VP16-UAS binary transgenic system	Smoa1/AKT1	Tg(UAS:*smoa1*-GFP; *krt4*:Gal4-V16)/Tg(UAS:*myrhAKT1*)	Brain tumor formation when genes are co-expressed	[121]
	Gal4VP16-UAS binary transgenic system	Smoa1	Tg(UAS:*smoa1*-GFP; *krt5*:Gal4-VP16)	Gliomagenesis driven by Shh activation	[122]
	TetOn (Doxycycline inducible)/Gal4VP16-UAS systems	KRAS	Tg(UAS:mCherry-KRAS^G12V^; *krt5*/*gfap*:Gal4-VP16)/Tg(TRE:mCherry-KRAS^G12V^; *krt5*/*gfap*:rtTa)	Malignant brain tumors driven by Ras and mTOR activation	[123]
	Gal4-UAS system	HRAS/YAP	Tg(UAS:GFP-HRAS^G12V^; *zic4*:Gal4-VP16)/Tg(UAS:YAP^S5A^)	Massive brain tumors and increased aggressiveness by YAP expression	[124]
	Tol2 system (tissue-specific promoter)	IDH1	Tg(*nestin*: eGFP-IDH1^wildtype^; IDH1^R132H^; IDH1^G70D^; IDH1^R132C^) Tg(*gfap*: eGFP-IDH1^wildtype^; IDH1^R132H^; IDH1^G70D^; IDH1^R132C^) Tg(*gata2*: eGFP-IDH1^wildtype^; IDH1^R132H^; IDH1^G70D^; IDH1^R132C^)	No brain tumor development, suggesting additional transformation events are required	[125]
	Gal4-UAS system	*ptf1a*/Rac1/Akt1	Tg(UAS:*myrAKT1*; *ptf1a*:Gal4-VP16)/ Tg(UAS:GFP-RAC1^G12V^; *ptf1a*:Gal4-VP16)	Importance of Akt1 in gliomagenesis and Rac1 in progression	[129]
	Tol2 (tissue-specific promoter)/LexPR transcriptional activator system	AKT1/*cxcr4*	pDEST-lexOP:AKT1/ pDEST-lexOP:AKT1/ *cxcr4b^-/-^* mutant	Tumor formation with increased microglia in neural cells by infiltration of peripheral macrophages via Sdf1b-Cxcr4b signaling	[130]
	Morpholinos	Ephrin-B3/EphA4	Tg(*fli*:EGFP)	Ephrin-B3 promotes angiogenesis by inhibition of EphA4-induced apoptosis	[127]
Knockdown	Morpholinos	PlexA1	Tg(*kdrl*:eGFP)	Abnormal angiogenesis. Potential prognosis marker	[128]
	Morpholinos	PJA1	WT	PJA1 possibly downregulates apoptosis, leading to gliomas	[131]
Knockout	ENU/ZFNs	*nf1a*/*nf1b*	Tg(*gfap*:GFP)/Tg(*sox10*:GFP)/Tg(*olig2*:GFP)/*p53^-/-^*	High grade gliomas and MPNSTs. Hyperactivation of ERK and mTOR pathways	[132]
	CRISPR/Cas9	*atrx*	WT/Tg(*gata1*:GFP)/*p53^−^*^/*−*^/*nf1^−/−^*	Development of various malignancies and downregulation of telomerase	[133]
Knockout/Transgenesis	Tol2/LexPR transcriptional activator system/CRISPR/Cas9	AKT1/*p2ry12*	pDEST-lexOP:AKT1	Microglia-tumor cell interactions are initiated by Ca2+-dependent ATP release from pre-neoplastic cells and their coupling with P2ry12	[134]
Overexpression	Human GLUD2 mRNA injection	GLUD2	WT	Impaired glial cell proliferation. Novel target for GBM progression	[135]

Dysregulation of cellular energetics and alteration of tumor cell metabolism, in particular glutamine metabolism, has been investigated using zebrafish models. For instance, glutamate dehydrogenase 2 (GLUD2) mRNA injection in zebrafish resulted in glial cell proliferation impairment while neuronal development was not affected, suggesting a novel potential drug target for GBM progression [135]. Additional studies combining the use of MOs and xenotransplantation in zebrafish for GBM research are reviewed in Section 4.3.

Other mutagenesis approaches used to generate stable zebrafish cancer models include: Targeting Induced Local Lesions in Genomes (TILLING), based on the exposure to the mutagen ethyl-nitrosourea (ENU), Zinc Finger Nucleases (ZFNs) and Transcription Activator-Like Effector Nucleases (TALENs), both based on nucleases and, most recently, CRISPR/Cas9 technology. These approaches have been used to demonstrate that NF1 loss increases penetrance of high-grade gliomas and malignant peripheral nerve sheath tumors in zebrafish with downregulation in telomerase and hyperactivation of ERK and mTOR pathways [133], as has also been observed in mice and human NF1-derived malignant peripheral nerve sheath tumors (MPNSTs) and gliomas [132].

A larval zebrafish model mimicking the earliest stages of brain tumor growth has been developed by overexpressing a cancer-promoting version of the human AKT1 oncogene in neural cells, under the neural-specific beta tubulin promoter, using the LexPR transcriptional activator system combined with macrophage and microglia reporters. In response to the activation of this oncogene, an increase in macrophage and microglia cell numbers was found in the brain, mediated by the SDF1b-CXCR4b signaling pathway which interacts with neoplastic cells by way of Ca^2+^-dependent ATP release and its coupling with the microglia P2ry12 receptor, inducing their proliferation. This infiltration did not result in enhanced phagocytic activity of the microglia towards the preneoplastic cells even though that is the primary role of the cells in the central nervous system [130,134]. Furthermore, macrophage depletion resulted in reduced proliferation of the tumor cells [130]. Along the same lines, in 2016, a zebrafish model of glioma with labeled macrophages/microglia demonstrated that microglia do not engulf or phagocytose glioma cells, but rather promote their growth [136]. It is worth pointing out that this effect has been attributed to overexpression of CD47 on glioma cells, which binds to SIRPα on the phagocytic cells to inhibit their phagocytic activity [10]. Accordingly, in patients, CD47 expression increases with the grade and is associated with lower survival rates [137]. These findings suggest the use of anti-CD47 therapy for the treatment of glioma to improve the ability of macrophages/microglia to fight against the tumor. 

As a whole, genetic zebrafish models have been successfully used to study brain tumors from their initial stages of development, offering the possibility to manipulate selected pathways relevant in glioma and immune-related cells (i.e., macrophages), to study their influence on the progression of the disease and also to look into the efficacy of their therapeutic intervention. 

### 4.3. Zebrafish Xenotransplantation Glioblastoma Models

Zebrafish xenografts commonly consist of the implantation of labeled cell lines of human or murine origin, or patient-derived cells into appropriate anatomical sites to generate in vivo heterotopic or orthotopic tumor models, which allow the tracking of a cancer cell’s survival, proliferation, metastasis, angiogenic potential, interaction with the microenvironment and response or resistance to new treatments [138,139,140]. The vast majority of zebrafish xenografts have been injected at 2 days post-fertilization (dpf), preventing tumor rejection, as the zebrafish adaptive immune system is not mature until 4–6 weeks post-fertilization, and during the first 12–14 days of development, only innate immune cells are present. [104,141]. Additionally, the transparency of the embryo at these early stages of development allows tumor cells to be tracked at a high resolution and the availability of reporter lines enables cell-host interactions to be inferred [140]. Although the injection site may depend on the cell type and the biological events to be interrogated, most implantations are performed in the yolk sac, the duct of Cuvier, the perivitelline space or the intraperitoneal cavity in the case of adult xenografts [113,139,142,143,144,145,146,147,148,149] (Figure 3). In the particular context of brain tumors, the likelihood of developing orthotopic tumors prompts most researchers to inject cells directly into the ventricles or the hindbrain-midbrain boundary. Although this is not optimal, it allows to study the growth of cancer cells in an environment which resembles the nervous system.

Xenograft models have provided information about GBM initiation and progression (Table 2) and have allowed investigations of novel therapeutic approaches (reviewed in Section 5). To investigate brain cancer biology, human far-red-labeled GBM cells were injected in zebrafish larvae to follow tumor progression, size, shape and brightness [150]. In this study, large numbers of tumor cells and cellular divisions over time were quantified by a combination of stereomicroscopy, light sheet fluorescence microscopy and flow cytometry. A robust, fast and automatable transplantation approach has recently been reported by Pudelko et al. to establish orthotopic GBM tumors in zebrafish. By injecting GBM cell lines or patient-derived samples into zebrafish blastulas, they observed their robust migration into the developing nervous system, already establishing an orthotopic intracranial tumor at 24 h post-injection (hpi). This approach avoids the technically challenging intracranial transplantation of single embryos, thus enabling the transplantation of hundreds of embryos per hour and providing an orthotopic vertebrate GBM model useful for drug discovery screens [151]. Others have used stem cells cultures derived from pediatric high-grade glioma tumors to generate orthotopic brain tumors, conserving stemness properties and resembling human gliomas [152]. The engraftment of patient-derived GBM cells in adult zebrafish has been assessed in an optically clear, immunocompromised homozygous compound mutant (*prkdc*^−/−^, *il2rga*^−/−^). Tumor cells robustly engrafted and proliferated showing similar growth kinetics and histopathology to those observed in a mouse model. Nevertheless, in this case, the requirement to pre-treat animals with clodronate liposomes in order to inhibit early macrophage ingestion might be too toxic, restricting the usefulness of this model [153]. The injection of glioma cells intracranially or in the yolk sac of 2 dpf embryos was used to demonstrate that chondroitin 4-sulfate (C4S) and chondroitin 6-sulfate (C6S) promote GBM cell migration and invasion [154]. The impact of nitric oxide (NO) in tumor progression was confirmed in 2 dpf Tg(*fli1*:EGFP) embryos injected in the yolk sac with rat GBM-labeled cells (GV1A1) [155]. Cells were able to grow and produce NO, as confirmed with diamino-fluoresceins and diamino-rhodamines, and the increase in the NO synthases (nos1 and nos2a). Furthermore, neovascularization and increase in *vegfa* and *cyclin D1* expression were observed in 85% of the embryos with NO production, and the addition of a NO scavenger (CPTIO) reduced *vegfa* and *cyclin D1* expression as well as the number of endothelial cells, suggesting that the reduction in NO levels by nitric oxide scavenging could be an efficient approach for the treatment of glioma. 

The mechanisms of glioma-vessel interactions, angiogenesis and tumor invasion have also been studied in zebrafish models. Umans et al. intracranially injected an adult glioma cell line and a pediatric patient derived xenoline into Tg(*fli1a*:eGFP) casper embryos at 3 dpf and observed the attachment of cells to the brain vasculature within 24 hpi, as well as their expansion and migration along the vascular network by 7 dpf [161]. The same cells injected into the trunk failed to interact with the vasculature and moved rostrally towards the brain, highlighting the importance of the microenvironment. Similarly, zebrafish xenograft models were used to elucidate the role of mesenchymal stems cells (MSCs) in GBM progression [160,164]. By co-injecting bone marrow-derived MSCs with two different GBM cell lines (U373 or U87) in the brain of zebrafish embryos, intratumor heterogeneity was observed as a consequence of the differential crosstalk between MSC-GBM cells, affecting invasion in a cell type-specific manner [160]. Such different tumor progression depending on the cell type and its interaction with microglia was demonstrated using a macrophage/microglia-labeled transgenic zebrafish line or the interferon regulatory factor 8 mutant (*irf8*^−/−^*),* which does not contain microglia [136]. The injection of U87 or U251 cells into the optic tectum resulted in tumor progression and the attraction of microglia in both cell types although interactions and infiltration differed in number and nature. The injection of cells in *irf8*^−/−^ decreased proliferation and survival, demonstrating the protumoral activity of microglia and how important it is to consider their differential response to distinct cell types when developing immunotherapies to treat gliomas. TGF-β was also shown to enhance tumor-induced angiogenesis via the JNK pathway and macrophage infiltration in a zebrafish glioma xenograft model [156]. Pretreatment of U87 GBM cells with TGF-β1 and subsequent injection into the yolk sac of Tg(*fli*:GFP) embryos at 2dpf led to a significant increase in newly-formed blood vessels, as well as in macrophage accumulation in the brain region, tail and yolk sac. Additionally, the treatment of xenografted embryos with a JNK inhibitor resulted in a decrease in angiogenesis, which may well be a powerful model for anti-angiogenesis drug screenings. The same group established a tumor invasion model in 2dpf Tg(*fli*:EGFP) embryos by transplanting the U87 cell line and its derived cancer stem cells (CSCs) [157]. CSCs enriched from U87 cells spread rapidly through the vessels, acquiring a protrusive appearance to establish metastasis and their invasiveness correlated with the up-regulation of the stem cell marker CD133 and the MMP9. Furthermore, CSC invasion was markedly inhibited by an MMP9 inhibitor, validating the use of zebrafish models to study the mechanisms underlying the invasion and metastatic behavior of glioma cells.

To examine the effects of Rac proteins on GBM progression in vivo, Lai et al. injected GFP-labeled U373- tumor sphere cells (GBM stem-like cells) in the yolk sac of 2 dpf Tg(*kdr*:mCherry) embryos with labeled endothelial cells, harboring control scramble short haipin (shRNA), Rac shRNAs (shRacs) and Rac 1–3 cDNAs, to silence or overexpress Rac proteins respectively [158]. They observed higher survival rates and lower incidence of angiogenesis in embryos bearing the tumor spheres derived from U373-MG cells with shRacs than controls, correlating the expression of Rac proteins with aggressiveness and poor prognosis in GBM. RECQ1 helicase plays an important role in tumor progression, as its expression is highly elevated in GBM [159]. shRNA-silencing of *RECQ1* in U87 cells transplanted into 2 dpf zebrafish brains resulted in decreased tumor growth. MOs were used to knock down lysine (K)-specific methyltransferase 2A (*KMT2A*), involved in glioma progression [162], resulting in downregulated proliferation of neural progenitors, premature differentiation of neurons and impaired gliogenesis [165]. KMT2A-knockdown in U87MG cells transplanted into 2 dpf Tg(*fli1*:EGFP) zebrafish brains increased both tumor growth and angiogenesis, even when embryos were treated with an immune suppressant (dexamethasone), revealing that KMT2A acts negatively on tumor growth in contrast with the previous conception of *KMT2A* as an oncogene in tumorigenesis [162]. Gamble et al. discovered that adhesion to laminin subunit alpha 5 (*lama5*), an important component of blood vessels, decreases GBM cell invasion and promotes the formation of blood vessel dependent microtumors. With 4D individual cell tracking technology and U251MG cells, xenotransplantation in the hindbrain ventricle of Tg(*fli1*:EGFP) 2 dpf embryos, proliferation, dispersion, microtumor formation and cell/blood vessel association were confirmed [163]. Nevertheless, *lama5* knockdown by MOs resulted in significantly higher cell dispersion and mobility whereas microtumor formation was lower. Thus, *lama5* increases GBM cell attachment to blood vessels by elevating VEGF activity, which in turn suppresses invasion but increases tumor formation.

Overall, a broad variety of xenograft zebrafish models of GBM have been established with different purposes. Although orthotopic models are preferable to better mimic the human pathology, heterotopic models allowing for the easier implantation of cells can be used for a preliminary screening of drugs and/or to study particular mechanisms.

## 5. Evaluation of New Treatments for GBM Using Zebrafish Models

GBM is a devastating disease, not only because of its particular aggressive nature, but also due to the very limited efficacy of the therapeutic options currently available [166,167]. The standard of care is surgery, followed by radiotherapy and TMZ, a DNA alkylating agent that can be administered orally [166,168]. Recurrence of GBM is very high and survival after treatment is commonly 12–15 months due to the following issues [166,167]: (i) difficulties to remove all tumors due to their great invasive and proliferative capacity [167,168], (ii) their high mutational capacity, which rapidly generates resistance to chemotherapies, such as TMZ [166,169]. In addition to therapies targeting cancer cells, novel anti-tumoral immunotherapies have been evaluated and numerous clinical trials using immune checkpoint inhibitors (ICIs, i.e., anti-PD1 and anti-CTLA4) are ongoing. Furthermore, therapeutic strategies to reprogram the tumor microenvironment or to normalize angiogenesis, including targeting and re-educating TAMs towards M1 anti-tumoral macrophages are of particular interest. For example, intratumoral delivery of IL-12 using a genetically-modified virus, alone or in combination with ICIs, showed good results in preclinical murine models [84]. For the genetic, epigenetic and metabolic reprogramming of GBM tumors, several approaches are being investigated. For instance, histone deacetylate inhibitors and shRNAs towards HDAC1 and 2 [170] or SIRT1, alone, or in combination with radiotherapy, showed antitumoral efficacy [171,172]. Either to kill cancer cells or to reprogram the TME, miRNA manipulation represents a very attractive target (reviewed in Section 3.4). Other major issues for the effective treatment of brain tumors are the ability of the drugs to cross the blood–brain–barrier (BBB) and/or the absence of systemic toxicity. In this scenario, zebrafish models have arisen as very useful pre-clinical models to evaluate in vivo the anti-tumoral efficacy of new drugs and their ability to reach and interact with target cells (i.e., cancer cells and macrophages), as well as to modulate the TME (i.e., angiogenesis) and their biodistribution ability to cross the blood–brain–barrier (BBB) and/or toxicity. A brief description of relevant studies using mainly zebrafish xenograft models for testing GBM therapies is provided below and in Table 3.

### 5.1. Treatments Tested in Zebrafish by Xenotransplantation of GBM Cell Lines

Zebrafish xenotransplants have allowed GBM growth and response to treatment to be studied in real time, which, in turn, enables an evaluation of the proliferative, migratory, invasive and angiogenic status of the tumor. In these studies, TMZ was commonly used as the reference treatment and for positive control. For example, 2-methoxy-6-acetyl-7methyljuglone (MAM), a natural product that induced necroptosis in colon and lung cancer cells, injected into U251-xenograft zebrafish models showed similar anti-tumoral efficacy to TMZ at very low doses [175]. A comparison between TMZ and bortezomib, a proteasome inhibitor, in GBM9-xenograft zebrafish, showed similar tumor reduction for both drugs but lower survival for bortezomib [168]. Using the same model, tumor regrowth after TMZ treatment was demonstrated for cancer stem cells expressing Sox2 and GFAP, revealing molecular mechanisms underlying resistance to treatment [169]. In another study, the VEGF and its coreceptor and proangiogenic factor neuropilin-1 (NRP-1) were knocked down in two human patient–derived GBM cell lines, GBM1A and GBM22, previously reported to be resistant to TMZ, before implantation into the brain ventricle region of 36 hpf embryos [173]. Although deletion of both VEGF and NRP-1 were able to inhibit tumor growth, TMZ treatment showed better results in combination with NRP-1 abrogation. Onalaspib, an inhibitor of heat shock protein 90 (HSP90), alone or in combination with TMZ, was tested in GBM zebrafish models that were injected in the mid-hindbrain with U251HF-GFP cells, showing better activity for the combination therapy [176]. Honokiol, a natural compound with anti-inflammatory, anti-microbial, anti-oxidative and anti-depressant properties, was tested for its anti-tumoral activity in zebrafish models using U87MG cells injected into the yolk sac. Honokiol prevented the migration of the cancer cells to the brain and tail, reduced migratory, invasive and proliferative tumor activity with an increase in caspase-2 and inhibited the EGFR, CD133, Nestin, STAT3 phosphorylation and AKT/ERK signaling pathways [177]. In the U87MG-zebrafish model, micellar-nanoparticles (NPs) loaded with honokiol and doxorubicin prevented tumor progression; in an embryonic angiogenesis-zebrafish model Tg(*flk1*: EGFP), the same NPs inhibited the growth of intersegmental vessels (ISVs) thanks to a controlled release of both drugs, in a more effective way than NPs loaded with just one of the treatments [178]. Zebrafish models also offer advantages in terms of ethical and economic issues for the screening of new treatments. After an initial in vitro selection of drugs from traditional Chinese medicine to treat GBM stem cells, the 13 best candidates were evaluated for toxicity and anti-tumoral efficacy in U87MG-zebrafish models, resulting in the selection of clofoctol as the compound with the best activity ascribed to overexpression of a pro-apoptotic factor (KLF13) [179].

Genetic and epigenetic therapies have been also tested using zebrafish GBM models. Protein arginine methyltransferase 5 (PRMT5) inhibitors, which regulate gene expression by methylation of histonic and non-histonic proteins, improved survival after microinjection of GBM human derived cells expressing GFP in the midbrain-hindbrain of 36hpf casper zebrafish [180]. Schnekenburger et al. synthetized and tested inhibitors of HDAC and sirtuin (SIRT) in GBM-xenografts, showing their efficacy in preventing tumor growth [181]. An interesting pharmacological approach has been based on the following observation by Pudelko et al.: human mutT homologue 1 (MTH1), an enzyme responsible for degrading oxidized nucleotides—which have high oxidative pressure and protect tumor cells—is up-regulated in GBM and related to a poor prognosis [184]. To treat this condition, MTH1 inhibitors, which force the cancer cell to incorporate oxidized nucleotides into the DNA, have been tested in orthotopic GBM-zebrafish models, showing satisfactory anti-tumoral efficacy. Tiek et al. studied in vitro the functions of the three isoforms of the estrogen-related receptor β (ERR-β) in the context of GBM. By using a TMZ-resistant GBM cell line labeled with DiL and microinjected into the brain of zebrafish, significant anti-tumoral activity was revealed when an ERR-β agonist (DY131, which displaces the ERR-β towards the ERR-β2 isoform) was combined with a CLK inhibitor (TG-003) [182]. In another study, 5-fluorouracil and erlotinib (tyrosin kinase inhibitor) reduced tumor size in zebrafish xenotransplants with murine-GBM-cells overexpressing ERR-β2 [183]. Thus, these treatments could be a good alternative for TMZ-resistant tumors.

Zebrafish models have been applied to evaluate the outcomes of metabolic manipulation in cancer progression and GBM [187]. Shtraizent et al. showed how mannose phosphate isomerase, as a metabolic enzyme, can maintain Warburg metabolism in zebrafish embryos with GBM [188]. Using the same zebrafish tumor model, Wehmas et al. observed significant tumor inhibition after treatment with LY294002, a selective PI3K inhibitor [167]. These results validate the same findings observed in organotypic mouse brain tissues [189].

### 5.2. Use of Zebrafish to Test Anti-Angiogenic Activity

Inhibition of angiogenesis is applied for the treatment of cancer to reduce the supply of oxygen and nutrients to the tumor, commonly in combination with other therapies [190]. Several zebrafish models have been used as tools for evaluating angiogenesis as well as brain pathologies in real time [191,192]. An interesting protocol to study tumor angiogenesis using zebrafish embryos was established in 2007 [139] and advantages/limitations of this model versus others have been reviewed by Nowak-Sliwinska et al. [193]. In the context of GBM, Wang et al. tested natural inhibitors of cyclooxygenase-2 (COX-2), microsomal prostaglandin E synthase-1 (mPGES-1) and cytochrome P450 (CYP4A11) as enzymes involved in angiogenic development [194]. Among different flavonoids, isoliquiritigenin (ISL) showed the most potent anti-angiogenic activity in zebrafish and rabbit corneal models. This anti-angiogenic effect led to an improved anti-tumoral efficacy and normalized glioma vasculature in combination with TMZ in murine models of glioma. In another study, the chemotherapeutic agent dianhydrogalactitol (DAG) was evaluated in zebrafish models for phase II clinical GBM trials where inhibition of migration and invasion of glioma cells was seen as well as dose-dependent reduction in the expression of VEGF, VEGFR2, FGF2 and FGFR2, all of which correlated with reduced tumor angiogenesis [195]. Bousseau et al. demonstrated the ability of a phosphite (PST3.1a) to block angiogenesis by interaction with VEGFR2 and galectin-1 [196], using zebrafish. Nordy, an inhibitor of arachinodate 5-lipoxygenase, was compared with well-known VEGF receptor tyrosin kinase inhibitors (i.e., Vetalanib, Suntinib and Axitinib) in zebrafish embryos whose yolk sacs had been injected with GSCs; the results showed how they were able to block tumor-induced vessel formation and inhibit the invasion and proliferation of GSCs by promoting their differentiation [186]. 

### 5.3. Use of Zebrafish to Test the Ability of Drugs to Cross the Blood–Brain–Barrier

A major limitation in the application of new drugs in the fight against GBM is their ability to cross the Blood–brain–barrier (BBB). This physiological barrier limits the passage of molecules from the circulatory system to the central nervous system (CNS) and is the cause of failure for 98% of the drugs that target the CNSs tested in clinical trials [197].

Previous studies on the development and maturation of the BBB in zebrafish have shown a sophisticated BBB, which is functionally and structurally similar to that of higher vertebrates. Although it is not exactly the same, these studies suggest the use of zebrafish as an experimental model organism for BBB-penetrating drug screenings [174,198]. Consequently, several drug delivery systems have been tested with the aim of improving the ability of pharmacological molecules to cross it. According to studies carried out on both mammals and zebrafish, their similar BBBs have been attributed not only to the neurovascular cellular composition, namely endothelial cells, pericytes, glia, neurons and microglia, but also to their associations, tight junction proteins and active transport systems [199]. Likewise, high concordance has been reported in the development and function of CNS capillaries and underlying molecular events driving these processes [192]. Additionally, the expression of Claudin-5 and ZO-1, which is concomitant with the maturation of the BBB, was detected in cerebral microvessels from 3 days dpf [174,198]. Nevertheless, several issues have been encountered in certain studies. For instance, zebrafish present a population of radial glia instead of the classic stellate astrocytes in mammals and although this glia expresses relevant astrocytic signals and also plays a role for ion homeostasis in the brain, some differences which remain to be fully understood might be encountered [200]. Some differences between mammalian and zebrafish pericytes have been found, such as lack of expression of canonical markers in the latter (i.e., Rgs5a or Desmin a/b) [201]. The origin of cranial pericytes is exclusive to the neural crest in mammals, while in zebrafish pericytes may also be of mesenchymal origin [202]. Despite these limitations, the flexibility, predictability and translational value of rigorous experiments of zebrafish for humans should be considered when using zebrafish BBB models for preclinical screenings of new therapies. Below we provide some examples.

Doxorubicin (DOX), a chemotherapeutic commonly used in cancer and unable to cross the BBB, is encapsulated in an apoferritin nanocage to target the transferrin receptor 1 (TfR1) overexpressed in brain endothelial cells and GBM cells. These NPs, labeled with the fluorescent dye Cy5.5, upon injection in the hearts of zebrafish are able to cross the BBB, so are an interesting strategy against GBM [197]. The same TfR1 ligand is used in carbon nitride dots, conjugated with gemcitabine and transferrin (CN-GM-Tf) NPs, showing strong anti-tumoral efficacy in GBM zebrafish models [203]. Another approach consists of vincristine sulfate in powder form added to low-density lipoprotein (LDL, expressed in the endothelial cells of the BBB and glioma cells) NPs modified with the T7 peptide TfR ligand. This dual interaction is also effective in crossing the BBB and acting on GBM, but using LDL nanocarriers could be problematic in patients with high cholesterol levels [204]. Zou et al. used monosialotetrahexosylganglioside (GM1) as a targeting ligand to improve the transport of DOX-loaded micelles through the BBB [205]. LysoGM1, a product of GM1 hydrolysis with a hydrophilic group to improve the transport across the BBB, is used to functionalize poly(lactic-co-glycolic acid) (PLGA) NPs loaded with DOX. DOX/GM1 and PLGA-lysoGM1/DOX are able to increase the anti-tumoral efficacy of DOX in GBM zebrafish models and improve nerve functions [205,206].

### 5.4. Use of Zebrafish to Test the Toxicity of Drugs

In addition to the studies detailed above, zebrafish have been widely used for toxicity, biocompatibility and/or biodistribution screening of drugs, benefiting from its ease of use and ethical and economic advantages versus other animal models. For example, cobalt (III) bound to nimesulide (Co-NMS), a COX-2 inhibitor, has been proven to increase the effects of radiotherapy in GBM and its neurotoxicity has been evaluated in zebrafish models from the changes in neurobehaviour capacity (swimming activity and movement) and morphological abnormalities in the CNS from hematoxylin/eosin staining of histological sections of the brain. Reduction in brain function and development has been observed at Co-NMS concentrations higher than 10 µM, while 5 µM concentration has been considered to be safe; there has also been a more intense radiotherapy effect on GBM in terms of a rise in the generation of ROSs and mitochondria damage in tumor cells [207]. Several alkaloids (i.e., moschamine, N-p-coumaroyl serotonin) have been tested in zebrafish, which showed no toxicity, although cytostatic and cytotoxic effects towards GBM cells were induced [208,209]. Curcumin, a neuroprotective phytotherapeutic, encapsulated in methoxy-poly(ethylene-glycol)-poly(Ɛ-caprolactone) (MPEG-PCL) NPs showed better absorption and biodistribution versus the free drug, in particular in lipophilic areas, such as the CNS and the yolk sac [210].

As a whole, several studies using zebrafish have been performed for large drug screenings and also to understand the mechanism of action of particular treatments in zebrafish GBM. Furthermore, optimized zebrafish models, described in Section 4, offer valuable tools for further studies, expected in the near future, on the effect of new drugs in particular cases (i.e., genetic modifications) and the TME (i.e., macrophages).

## 6. Conclusions

In recent years, seminal studies on glioblastoma (GBM) have unraveled the interactions between tumor cells and immune cells from the initial steps and along the progression of the pathology. In this regard, macrophages/microglia, representing up to 30-50% of cells in some tumors, present protumoral properties and functions which can, in theory, be reprogrammed with appropriate treatments, not yet available in the clinic, as suggested by some pre-clinical results described in Section 2. In fact, 11 open phase III clinical trials for GBM patients are ongoing, mostly related with immunotherapy [211]. Despite significant scientific efforts, the standard of care treatment for GBM is still surgery, followed by radiotherapy and TMZ, with only evaluation of IDH mutations and methylation of MGMT as molecular markers of routine to classify and treat the disease. Numerous studies have revealed genetic, epigenetic and metabolic alterations encountered in GBM. Among them, we highlight the identification of miRNAs and their changes in GBM as very attractive targets for diagnosis and/or therapeutic purposes, following a similar trend to other types of cancer [59,60,61,212].

In this context, genetic and xenotransplant zebrafish models have been successfully used to generate new knowledge about GBM pathology and some studies have been initiated for screening of new treatments. While there is still some room for improvement related to the modeling of the human immune system and optimization of orthotopic models of GBM in zebrafish, satisfactory results have already been established for detailed understanding of gene and metabolic alterations of the disease. 

Overall, we expect that zebrafish models will be further exploited by taking into account their biological, ethical and economic advantages for the screening and comprehensive evaluation of new drugs to ultimately improve the treatment of patients with glioblastoma.

## Figures and Tables

**Figure 1 cancers-13-01087-f001:**
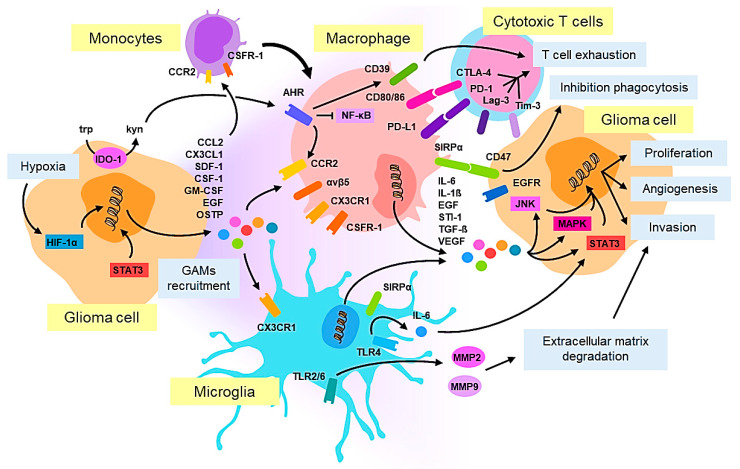
Representation of interactions between glioma cells, macrophages/microglia and other components of the tumor microenvironment. Macrophages and microglia (GAMs) are attracted to the tumor by several glioma cells-derived factors such as CCL2, CX3CL1, SDF-1, CSF-1, GM-CSF, EGF or OSTP, expressed in response to environmental stress (such as hypoxia) or GAMs-derived factors themselves. In the tumor microenvironment, GAMs exert a tumor-supporting activity through secretion of factors such as IL-6 (particularly in response to TLR4 activation in microglia), IL-1β, EGF, STI-1, TGF-β or VEGF that activate different signaling pathways (such as JNK, MAPK or STAT3 in glioma cells) promoting proliferation, angiogenesis or invasion of tumor cells. In microglia, TLR2/6 activation has been shown to induce MMP2 and MMP9 expression, contributing to extracellular matrix degradation and tumor invasiveness. GAMs are also immunosuppressive effectors, especially via the expression of molecules which lead to T-cell dysfunction (CD80/86, PD-L1 or CD39). The phagocytic activity of GAMs is also impaired by the expression of CD47 (binding to SIRPα) by glioma cells.

**Figure 2 cancers-13-01087-f002:**
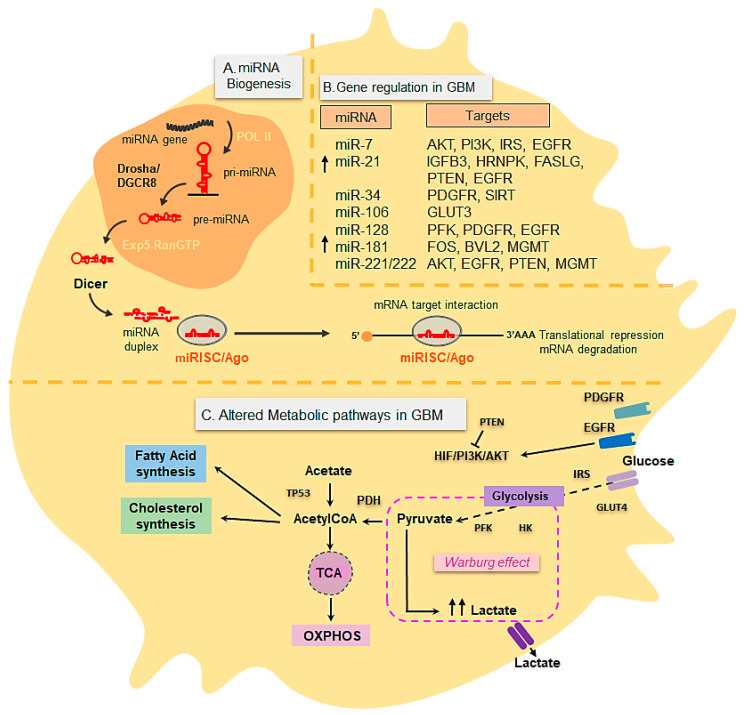
Representation molecular mechanisms underlying the tumoral properties of glioma cells. A. Canonical pathway for miRNA biogenesis. miRNA are transcribed by RNA Pol II to produce the pri-miRNA that is later processed by the endonuclease Drosha/DGCR8 complex resulting in the formation of the hairpin precursor (pre-miRNA) that goes under a second cleavage by the endonuclease Dicer and generates a miRNA duplex. One of the strains, the mature ~22 nt (guide strain) is later loaded into the RISC (RNA Induced Silencing Complex) in association with Ago proteins. Via imperfect base-pairing to the 3′ UTR of its target mRNAs, miRNAs regulate the expression of its target genes by repressing mRNA translation and/or promoting mRNA degradation. B. miRNAs and target genes involved in GBM. Examples of miRNAs found to be downregulated or upregulated (indicated with the arrow) in GBM and their target genes involved in GBM pathology. C. Metabolic pathways altered in GBM. Metabolic switching towards aerobic glycolisis is a hallmark of cancer which controls the properties and functions of tumor cells, such as their increased proliferation or resistance to treatments. The therapeutic intervention of metabolic pathways, described in this figure, may offer new opportunities for the treatment of GBM.

**Figure 3 cancers-13-01087-f003:**
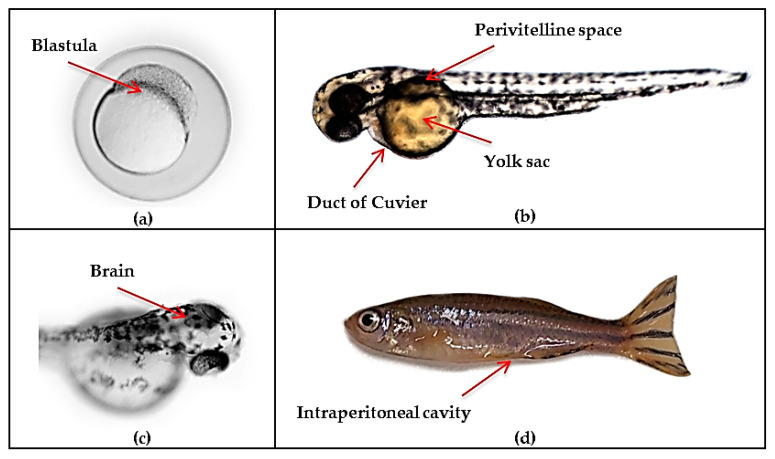
Injection sites commonly used in zebrafish xenotransplantation assays: (**a**) Blastula (3.5 hpf); (**b**) Duct of Cuvier, perivitelline space and yolk sac in zebrafish embryo (48 hpf); (**c**) Embryo brain (48 hpf); (**d**) Intraperitoneal cavity in adult zebrafish. Images (**a**–**c**) were acquired in our lab with a stereomicroscope (AZ-100 Multizoom, Nikon). Image (**d**) was acquired in our lab with a Samsung Galaxy A70.

**Table 2 cancers-13-01087-t002:** Xenograft approaches using zebrafish for modeling GBM.

Injection Site	Cell Line	Stage	Zebrafish Strain	Notable Results	Reference
Yolk sac	Far-red CCF-STTG1	48 hpf	WT	Tumor progression size, shape, brightness and quantification of tumor cells, by combination of LSFM and flow cytometry	[150]
	GV1A1-CM-DiI	48 hpf	Tg(*fli1*:EGFP)	Impact of NO production via *vegfa* and *cyclin D1* expression	[155]
	U87-RFP (TGF-β1-treated)	48 hpf	Tg(*fli*:GFP)	Increase in newly-formed blood vessels and macrophage accumulation in the brain region	[156]
	U87-RPF (CSCs-enriched cells)	48 hpf	Tg(*fli*:GFP)	Up-regulation of CD133 and MMP9 leads to glioma invasiveness	[157]
	U373-GFP (GBM stem-like cells)	48 hpf	Tg(*kdr*:mCherry)	RAC proteins promote aggressiveness and poor prognosis of GBM	[158]
Yolk sac/ Brain	U251- CM-DiI	48 hpf	WT	C4S and C6S promote GBM cell migration and invasion	[154]
-	Patient-derived/ U343-MGA-GFP/ GBM primary cultures	Blastula (3.5 hpf)	Tg(*fli1a*:EGFP)/Tg(*elavl3*:GFP)/Tg(*mpeg1*:mCherry)	Novel transplantation procedure with development of orthotopic intracranial tumors and macrophage/microglia interactions	[151]
Ventricles	BPC-A7-RFP (stem cell cultures derived from pediatric brain tumors)	48 hpf	WT	Initiation of glioma-like tumors from stem cell cultures, conserving stemness properties	[152]
Optic tectum	U87-mCherry/U251-mChery	72 hpf	Tg(*mpeg1*:EGFP)/*irf8*^−/−^)	Microglia protumoral activity and differential response to particular cell types	[136]
Brain	*RECQ1*- silenced U87-DsRed	52 hpf	WT	RECQ1 plays an important role in tumor progression. Promising approach for GBM treatment	[159]
	DiI/DiO- Bone marrow -derived MSCs/U373-eGFP/U87-dsRED	52 hpf	WT	MSC-GBM cell crosstalk affects invasion in a cell type-specific manner	[160]
	D54-MG/D2159MG	72 hpf	Tg(*fli1a*:eGFP); casper	Cell attachment and migration through the brain vasculature. Importance of microenvironment	[161]
	*KMT2A*- knockdown U87MG	48 hpf	Tg(*fli1*:EGFP)	Increase in tumor growth and angiogenesis. KMT2A acts negatively on tumor growth	[162]
Hindbrain ventricle	U251MG in *lama5* knockdown embryos	48 hpf	Tg(*fli*:EGGP)	*lama5* suppresses invasion but increases tumor formation via VEGF	[163]
Intraperitoneal	Patient-derived	Adult	Casper, *prkdc^−/−^*, *il2rga^−/−^*	Successful tumor engraftment at physiological temperature (37 °C)	[146]

**Table 3 cancers-13-01087-t003:** Treatments for GBM evaluated using xenograft zebrafish models.

Treatment	Zebrafish Strain	Cell Line	Stage	Injection Site	Remarkable Results	Reference
TMZ	Casper	shNRP-1 or shVEGF-GBM1A and GBM22	36 hpf	Brain ventricle	TMZ enhances survival and decreases tumor growth. NRP-1 abrogation improves the effect of TMZ	[173]
TNB	Tg(*flk*:eGFP)- Casper	U87-RFP or U251-RFP	72 hpf	Brain	TNB is able to cross the BBB and inhibits tumor progression	[174]
MAM	WT	U251- DiI	48 hpf	Yolk sac	Inhibition of tumor growth, possibly in an apoptosis-independent manner	[175]
TMZ/Bortezomib	Casper	GBM9-GFP neurospheres and X12-v2	36 hpf	MHB	Diverse differentiation patterns in cells, but both positive for Sox2 and responsive to therapeutics	[168]
TMZ	Casper	GBM9-GFP neurospheres	36 hpf	MHB	Putative GBM stem cells are more resistant and might contribute to tumor regrowth	[169]
TMZ/Onalespib	WT	U251HF-GFP	36 hpf	MHB	Combination of Onalespib with TMZ reduces tumor burden and extends survival	[176]
HK	WT	U87MG-CM-DiI	48 hpf	Yolk sac	Inhibition of tumor growth and metastasis	[177]
Dox-HK-MPEG-PCL micelles	Tg (*flk*: eGFP)/WT	U87- GFP	14 and 48 hpf	Perivitelline space	Anti-angiogenic and anti-tumor properties	[178]
Clofoctol	WT	U87MG- SLCs-GFP	48 hpf	Yolk sac	Tumor inhibition without toxicity	[179]
PRMT5 inhibitors (CMP12, CMP5, HLCL65, HLCL66)	Casper	Patient-derived neurospheres (GBMNS-30–GFP)	36 hpf	MHB	Anti-tumor efficacy of CMP5, without toxicity	[180]
HDAC class III/SIRT1/2	WT	Hs683 and U373-CM-DiI	36 hpf	Yolk sac	Ability to abrogate tumor development	[181]
ERR-β agonists (TG-003/DY131)	Tg(*kdrl*:GRCFP)zn1; *mitfa^b692/b692^*; *ednr^b1b140/b140^*	42MGBA-TMZres-DiI	36 hpf	Intracranial	Shift to ERRb2 isoform and suppression of growth and migration in TMZ-resistant cells	[182]
5-FU/Erlotinib	Tg(*fli1*:eGFP)	Conditioned GBMERBB2-RFP	30 dpf	Cerebrum (intranasally)	Mouse brain tumors can grow orthotopically in fish and are responsive to treatment	[183]
MTH1 inhibitor (TH1579)	WT	CD33^+^ enriched fraction of patient-derived CMV-LUC/U343-MGA:GFP		Intracranial	Real-time death of glioma stem cells (GSCs) and tumor volume decrease	[184]
ZnO NP/LY294002	Tg(*fli1*:eGFP)	U87MG-CM- DiI	48 or 72 hpf	Hindbrain	ZnO NPs enhance cancer cell proliferation	[167]
C60 fullerene derivatives	Tg(*fli*:eGFP)	C6-PKH26 (murine neural stem cell)	24 hpf	Brain	Reduction in GBM formation	[185]
Axitinib, Suntinib, Vatalani/Nordy	Tg (*fli1*:EGFP)	GSCs U87- derived	48 hpf	Yolk sac	Inhibition of tumor-induced vessel formation. Model for anti-GSC drug evaluation	[186]

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
