# Peer review of "Cellular and Molecular Mechanisms Underlying Glioblastoma and Zebrafish Models for the Discovery of New Treatments"

_cancers, 2021, doi:10.3390/cancers13051087_

Round 1

Reviewer 1 Report

This review article addresses genetic, epigenetic, metabolic, and microenvironmental contributors to glioblastoma (GBM) in humans and the role for zebrafish models in exploring these contributors to GBM initiation and progression. Overall the article is interesting, well-researched, and well-cited, and most sections follow a logical flow. The article should be carefully reviewed for a moderate number of typographical errors and for correction of grammar. Specific recommendations are outlined below.

Section 1. Introduction

  1. Glioblastoma is typically abbreviated as GBM, not GB; this should be corrected throughout the manuscript for clarity
  2. The authors site incidence rates for GBM (line 57-58) but it is unclear what population is described – global incidence versus a particular country?

Section 2. Cellular pathology and tumor microenvironment…

  1. Discussion of cellular origin of GBM is completely lacking. It is understood that a single cell of origin has not been identified, but this section should be significantly expanded.
  2. This section is rather poorly written in comparison to the remainder of the manuscript and should be revised. Multiple sentences are difficult to understand and several of these are noted below.
    1. Lines 109-112 confusing, cannot follow the discussion points here
    2. Lines 155-158 confusing and does not match what is shown in Fig 1 (e.g., receptor expression described in text does not match what is shown in the figure)
    3. Lines 204-206 – are CD204 and IBA-1 known to be expressed by GAMs derived from either microglia or monocytes?
    4. Lines 228-229 – states GAMs support GB progression although prior paragraphs suggest this is not always the case
  3. There is a lot of confusing variability in the way that monocytes/macrophages/microglia are discussed in this section. Please edit for consistency and clarity as this is clearly an important cellular contributor to GBM.

Section 3.3 Metabolic changes in GBM

  1. Lines 318-320 this sentence seems to contain typos, it is unclear what is being described

Section 4.1

  1. Lines 472-474, lines 481-489: these sentences are far too long and very confusing.
  2. Line 501 – not sure that use of gills as a model for lung is widely accepted or accurate
  3. The discussion of how the zebrafish embryonic brain differs from human brain is inadequate – it is not so simple, nor so similar, as the authors describe. This section should be revised (lines 501-507).

Section 4.2 Genetic zebrafish models of GBM

  1. Lines 529-530 the fli1-EGFP line is not a reporter for macrophages
  2. Lines 533-552. Several zebrafish models use the Gal4/UAS system. Readers who have limited or no familiarity with zebrafish models are not likely to know what this is. Please explain. Multiple promoters are referenced (krt5, zic4, gfap) – please define for clarity what cell types express these genes in zebrafish.
  3. Lines 562-565 the description of PlexinA1 is quite confusing and does not make sense.
  4. No discussion of study of epigenetic contributors to GBM in zebrafish? If not, should be stated.

Section 4.3 Zebrafish xenotransplant

  1. Line 616 time point at which adaptive immunity develops is inconsistent to what was previously described in the manuscript, please reconcile.
  2. Line 622 zebrafish do not have a peritoneal cavity because they do not have a diaphragm. Zebrafish have a coelomic cavity. Please use appropriate anatomical terms.
  3. Line 624-625 growth of cancer cells transplanted into the brain ventricle cannot be equated to growth of cancer cells in the brain. Intraventricular growth of human brain cancers is rare. Transplant into the neuropil is more similar to the native environment. This statement should be modified.
  4. Consider a paragraph on xenotransplant methods, pros and cons, etc. to accompany Figure 3 before describing specific xenograft models.

Section 5 Evaluation of new treatments

  1. Line 738 – use of consistent terminology – previously referred to GAMs, now referring to TAMs?
  2. Line 743 – what does it mean that the described treatment was “satisfactory”?

Section 5.2 Use of zebrafish…anti-angiogenic activity

  1. How comparable is embryo angiogenesis to tumor angiogenesis? Similarities and differences? This would assist readers in evaluating the usefulness of the model.

Section 5.3 Use of zebrafish…BBB

  1. Line 848 – It is not clear that the BBB in 3dpf zebrafish is considered to be a mature BBB resembling that of adult mammals. Please delve into the literature a bit more here and clarify as needed. This is an important point given the described use of zebrafish embryos in testing drugs.

Section 6 Conclusions

  1. It is a bit unclear why the conclusion is so heavily focused on MP/microglia considering that this one of numerous topics discussed. Please revise.

Figures

  1. Figure 1 – please see comments in Section 2 regarding consistency between text and Figure 1. The T cells in Fig. 1 should be identified as a subtype or multiple T cell types should be displayed since their function is highly variable.
  2. Figure 2 – The section showing IDH-wt versus IDH-mut in part C is confusing, it is unclear what is intended by the relative position of the text compared to the associated pathway. “Pyruvate” is misspelled in part C. It is totally unclear why a portion of the PTEN pathway and PDGFR and EGFR are shown in this figure – their contributions should be shown or they should be removed.
  3. Figure 3 – Please correct “intraperitoneal” as described above.

Tables

  1. Generally the tables are well organized and informative.
  2. Table 2 – the injection sites are quite confusing between studies. What is the difference between intracranially, ventricles, optic tectum, brain? Please provide consistency. Please correct “intraperitoneal” as described above.
  3. Table 3 – this is very dense and hard to follow. Consider including either including a column that indicates what the treatment is targeting (e.g. angiogenesis) or subdividing the table. Is it necessary to include the zebrafish strain? Please be consistent in Stage (hpf versus dpf). The table title should be revised to indicate that these studies are performed in xenotransplanted zebrafish.

Author Response

Dear Reviewer,

Reviewer 2 Report

The manuscript entitled “Cellular and molecular mechanisms underlying glioblastoma and zebrafish models for the discovery of new treatments” by Pedro Reimunde reviews the cellular and molecular alterations involved in glioblastoma initiation and progression. The authors also describe a remarkable variety of approaches on how zebrafish could improve our knowledge of the disease and how it could participate to the evaluation of novel treatments.

The manuscript is clear, well presented and written as well as largely documented.

Minor comments:

  1. As zebrafish is used to test anti-glioblastoma drugs (paragraph 5.1), anti-angiogenic drugs (paragraph 5.2), the ability of drugs to cross the blood-brain-barrier (paragraph 5.3) or drug toxicity (paragraph 5.4), I would suggest the authors to write a statement about drug metabolism in zebrafish liver as compared to human drug metabolism.
  2. In Table 1, the authors present the Tol2 system like the Gal4/UAS, LexPR or TetON expression strategies. However, Tol2 allows the integration of the transgenes but not their expression and should not be on put on the same level as the expression systems. A term like “tissue-specific promoters” would certainly be more appropriate.
  3. Page 13, Line 530. The Tg(fli1:EGFP) line is mainly used to label endothelial cells and the vascular system, but not macrophages.
  4. Line 699. The authors should specify the Tg(kdr:mCherry) transgenic model.

Author Response

Dear Reviewer,
